# Application of New Hyperspectral Sensors in the Remote Sensing of Aquatic Ecosystem Health: Exploiting PRISMA and DESIS for Four Italian Lakes

Mariano Bresciani *, Claudia Giardino, Alice Fabbretto, Andrea Pellegrino, Salvatore Mangano, Gary Free and Monica Pinardi

Institute of Electromagnetic Sensing of the Environment, National Research Council of Italy (CNR-IREA), Via Bassini 15, 20133 Milan, Italy; giardino.c@irea.cnr.it (C.G.); fabbretto.a@irea.cnr.it (A.F.); pellegrino.a@irea.cnr.it (A.P.); mangano.s@irea.cnr.it (S.M.); free.g@irea.cnr.it (G.F.); pinardi.m@irea.cnr.it (M.P.)
* Correspondence: bresciani.m@irea.cnr.it

**Abstract:** The monitoring of water bio-physical parameters and the management of aquatic ecosystems are crucial to cope with the current state of inland water degradation. Not only does water quality monitoring support management decision making, it also provides vital insights to better understand changing structural and functional lake processes. Remote sensing has been widely recognized as an essential integrating technique for water quality monitoring, thanks to its capabilities to utilize both historical archive data for thousands of lakes as well as near-real time observations at multiple scales. To date, most of the applications developed for inland water have been based on multispectral and mid to coarse spatial resolution satellites, while a new generation of spaceborne imaging spectroscopy is now available, and future missions are under development. This review aims to present the exploitation of data gathered from two currently orbiting hyperspectral sensors (i.e., PRISMA and DESIS) to retrieve water quality parameters across different aquatic ecosystems, encompassing deep clear lakes and river dammed reservoirs.

**Keywords:** satellite images; hyperspectral data; water quality; PRISMA; DESIS; turbidity; chlorophyll-a; aquatic vegetation

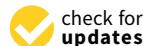

## 1. Introduction

Inland water ecosystems are crucial environments as they provide multiple ecosystem services and are vital resources for drinking, irrigation, sanitation, industry and recreation [1,2]. Many regions of the world are now facing serious challenges to their freshwater resource including increasing water demand, water shortage, decreases in the level of groundwater, reduced water quality and increasing damages to dependent ecosystems. Overall, in the past decades and under the current global warming scenario the quality of inland waters has been severely endangered and often dramatically degraded [3–6]. An expected decline in the quantity of water available is likely to coincide with an increasing demand for freshwater globally. To understand and anticipate this process in order to mitigate the deterioration of inland water quality, the monitoring of bio-physical variables depicting water conditions is essential. Water quality monitoring actually supports water authorities to take decisions, while it provides useful insights to lakes scientists to understand processes.

In situ measurements have been used for water quality monitoring for many years allowing the gathering of information from the entire water column, including an extensive set of parameters (e.g., phytoplankton abundance, nutrient concentrations) many of which eventually became parameters subject to national and international standards (e.g., [7–9]). For more than three decades, remote sensing has been recognized as an essential integrating technique for water quality monitoring. Through acquiring timely, frequent synoptic water

quality parameters, it also supports implementation of water protection directives [10], while it is a unique tool for a variety of scientific studies from local to global scale [11]. Satellite Earth Observation (EO) techniques are often used alongside in situ observations due to the possibility of having both historical archive data for thousands of lakes (since the 1970s) [12–14] and near-real time observational capabilities at multiple-scales.

Overall, the monitoring of water quality parameters in different types of waterbodies (lakes, rivers, etc.) can be optimally performed integrating traditional in-situ measurements with remote sensing and GIS techniques [15]. Water managers, in order to effectively deal with waterbodies ecological stressors, look to remote sensing as a useful tool to provide advanced spatial–temporal environmental information to support decision making [16–18]. An increasing demand for routine monitoring using remote sensing to support inland waters management to achieve water quality goals is now evident. Despite this recognized need and the numerous examples reported in the literature, water quality products derived by remote sensing are yet to be adopted on a routine framework, probably due also to the knowledge gap of water managers and policy makers to understand technical abilities and limitations of the different optical sensors [15]. For this reason, it is important that researchers and remote sensing experts communicate to water managers the appropriate available tools to address monitoring requirements [19–24]. To this aim international initiatives have been developed; among others GEO AquaWatch has several projects advancing in the coming years for promoting the use of scalable remote sensing and in situ water quality data. A key goal is to bridge the information gap between global decision-makers and water quality earth observation data providers.

To date, most of the applications developed for inland water quality monitoring and management have been based on multispectral and mid (e.g., Landsat constellation, Sentinel-2-MSI) to coarse (e.g., ENVISAT-MERIS, Sentinel-3-OLCI) spatial resolution satellites [25]. While research efforts are still ongoing to face the challenges typical of EO of inland waters, such as global chlorophyll-a concentration mapping or corrections for adjacency effects [26–30], it is also true that depending on an optical sensor's specifications, the EO-derived products vary from turbidity, transparency, and concentrations of chlorophyll, suspended particulate matter and colored dissolved organic matter, floating materials and, in the case of shallow waters, bottom depth and type. In the literature of the past and recent years, numerous articles and reviews [31] and reference therein addressed the optical water quality parameters that can be retrieved by remote sensing techniques (i.e., suspended sediments (turbidity), chlorophyll and other secondary pigments, color dissolved organic matter (CDOM), water clarity and temperature) (e.g., [13,32,33]), the different properties of sensors and platforms and their environmental applications (e.g., [34–36]) and the algorithms developed and implemented to retrieve water quality products (e.g., [37–39]). For the ecological analysis and correct management of lake ecosystems, it is of great importance to also have information on the benthic component of coastal areas and the presence of aquatic macrophytes. Most of the applications have been carried out with medium to high spatial resolution sensors (e.g., Landsat, Sentinel-2 MSI) supporting retrospective analysis, thematic mapping of species in valuable ecosystems such as wetlands.

In a recent review, Topp et al. [40] made a bibliometric analysis which highlighted the conspicuous increase of inland water quality remote sensing studies, particularly in the last 10–15 years studies were found to have moved beyond methods development towards water quality dynamics (e.g., trends, ecological and anthropogenic drivers, impacts of changing water quality on ecosystem function and water resources) thanks to an improved access to EO data and growing computational capabilities. In this review, the authors classified a subset of 236 articles into: Purely methodological, Methodological with pattern analysis, Trend/pattern analysis, and Water quality science research with a focus on impacts and drivers. In this latter more applied paper category a total of 40 articles were found, and the majority of them were related to lakes (85% of the applied articles). In addition, imaging spectroscopy has been also attracting a wide interest [28] and reference therein since its simultaneous collection of narrower and contiguous bands is improving aquatic ecosys-

tem mapping for the retrieval of parameters describing water quality, aquatic vegetation (e.g., biomass [41], invasive species identification [42]) and benthic substrates that might be undetectable with broadband multispectral sensors [15,27,40,43–48]. In such a context, airborne data (e.g., APEX, AISA, MIVIS) have been providing unique data at high spectral and spatial resolution for performing advanced mapping as well to support satellite mission development and verification (e.g., [49–52]). With respect to spaceborne observations, following the launch of Hyperion which was followed a few years later by Chris-PROBA and HICO, a new generation of spaceborne hyperspectral sensors (e.g., PRISMA, GaoFen-5, DESIS, HISUI), is now available for improving water resources monitoring, while future missions are also under development (e.g., EnMap, CHIME, SBG). Accordingly, this study aims to present the exploitation of data gathered from two currently orbiting spaceborne imaging spectrometers, PRecursore IperSpettrale della Missione Applicativa (PRISMA) and the DLR Earth Sensing Imaging Spectrometer (DESIS), for water quality parameters retrieval, bottom type mapping and monitoring emergent macrophytes in four Italian water bodies, encompassing deep clear lakes, turbid shallow lakes and river dammed reservoirs.

## 2. PRISMA and DESIS Missions

PRISMA, a mission fully funded by the Italian Space Agency (ASI), is an EO system with innovative, electro-optical instrumentation that combines a hyperspectral sensor with a medium-resolution panchromatic camera [53]. The PRISMA orbit is characterized by a repeat cycle of approximately 29 days. The PRISMA payload is composed by an imaging spectrometer, able to take images at 30 m resolution in a continuum of spectral bands ranging from 400 to 2500 nm, and a 5 m resolution panchromatic camera. The PRISMA system provides the capability to acquire, downlink and archive images of all hyperspectral/panchromatic channels totaling 200,000 km² daily over the primary area of interest defined as: longitude in the range 180° W–180° E; Latitude: in the range 70° S–70° N. Although the revisit time in nadir-looking configuration is of 29 days, the system is capable of acquiring images distant 1000 km in a single pass (with a total rotation left to right side looking and vice versa) so that the temporal resolution can be significantly improved.

DESIS, the other imaging spectrometry system used in this study, is a hyperspectral instrument integrated in the Multi-User-System for Earth Sensing (MUSES) platform installed on the International Space Station (ISS). The mission is operated by Teledyne Brown Engineering, Alabama, USA, and the German Aerospace Center (DLR), Germany. DESIS is realized as a pushbroom imaging spectrometer spectrally sensitive over the VNIR range from 400 to 1000 nm with a spectral sampling distance of 2.55 nm. The main specifications of DESIS are shown in Table 1. The launch of DESIS to ISS was on 29 June, 2018. The commissioning and validation phases were finished in March 2019. DESIS is now working operationally and will continue until at least the end of 2023 [54].

**Table 1.** Main specifications of PRISMA (PRISMA Products Specification Document, 2019) and DESIS (from DESIS link https://www.dlr.de/eoc/desktopdefault.aspx/tabid-13614 (accessed on 20 October 2021)) instruments.

|  | PRISMA | DESIS |
| --- | --- | --- |
| Launch | 22 March 2019 | 29 June 2018 |
| Coverage | 70° N to 70° S | 55° N to 52° S |
| Target lifetime | 5 years | 2018–2023 |
| Orbit | SSO 615 km 10:30 LTDN | MUSES platform on ISS |
| Number of bands | VNIR: 66 (440–1010 nm), SWIR: 174 (920–2505 nm), PAN: 1 (400–700 nm) | 235 (no binning), 60 (binning 4) |
| Spectral sampling interval | VNIR: 7.2–11 nm, SWIR: 6.5–11 nm | 2.55 nm (no binning), 10.2 nm (binning 4) |
| Spectral coverage | 440 nm to 2505 nm | 402 nm to 1000 nm |
| Ground sampling distance | Hyperspectral: 30 m, PAN: 5 m | 30 m |
| Signal-to-noise ratio | >160:1 VNIR, >100:1 SWIR, >240:1 PAN | 195 (w/o binning), 386 (4 binning) (based on laboratory calibration) (albedo 0.3 @ 550 nm) |
| Radiometric resolution | 12 bits | 13 bits + 1 bit gain |
| Swath | 30 km | 30 km |

## 3. Study Area, Imagery Data and Processing

For the purposes of this study, a set of PRISMA and DESIS images have been explored over four inland water systems located in Italy as shown in Figure 1. We have selected the case studies in order to include lakes of different trophic status, depth profiles and different hydrogeochemical and ecological processes.

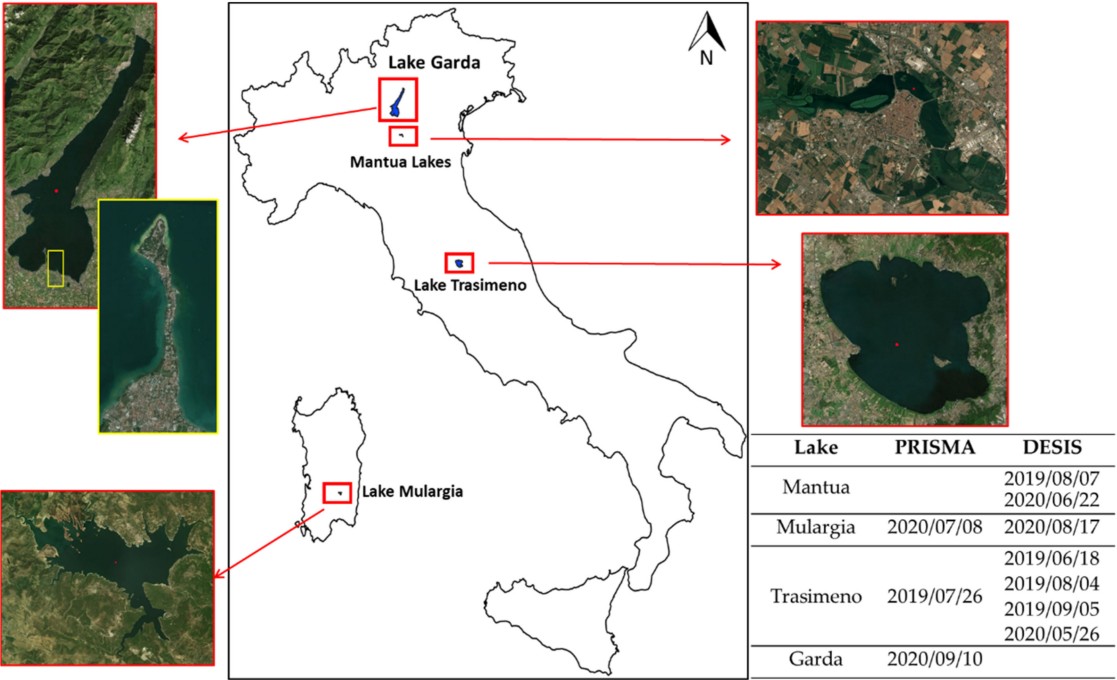

| Lake | PRISMA | DESIS |
|---|---|---|
| Mantua | | 2019/08/07<br>2020/06/22 |
| Mulargia | 2020/07/08 | 2020/08/17 |
| Trasimeno | 2019/07/26 | 2019/06/18<br>2019/08/04<br>2019/09/05<br>2020/05/26 |
| Garda | 2020/09/10 | |

**Figure 1.** Maps of the lake case studies investigated in Italy. The table reports the dates of PRISMA and DESIS images acquired for Mantua, Mulargia, Trasimeno and Garda lakes.

RISMA and DESIS, the latter being the binned product, have been firstly corrected for atmospheric effects with the ATCOR tool (version 9.3.0) [55]. These instruments have already proved capable of providing valuable atmospherically corrected reflectance data for water quality applications [56,57], while further research efforts to improve the retrieval of water reflectance are still under development [58]. The result of atmospheric correction is presented in Figure 2; both PRISMA and DESIS show distinct features associated with different gradients of trophic status, water transparency, bottom properties, in addition to emerging aquatic vegetation.

PRISMA and DESIS data corrected for the atmospheric effects were then turned into aquatic ecosystems products by inverting a bio-optical modelling to both map water quality parameters and bottom types while, vegetation indexes, were used to estimate the macrophyte biomass. In particular, the bio-optical model used in this study is similar to previously published case-2 water models (e.g., [59,60]), where water reflectance is depending on concentrations of water quality parameters; then, in case of shallow waters, also on bottom depths and on a fractional cover of bottom reflectances. The bio-optical model parametrization and inversion were run with the BOMBER code, detailed described in Giardino et al. [61] while Table 2 and Figure 3 show the site-specific parameters which allowed the bio-optical model implemented in BOMBER to be calibrated. The parameters for the four sites, which are resulting from previous field measurements (e.g., [27,62,63]), are comparable to literature data for case-2 waters (e.g., [64–66]). For the specific application over shallow waters, the bottom reflectance used in this study can be found in Ghirardi et al. [67]. Finally, the inversion procedure was initialized with the average values of water quality parameters typical of each site, for the season for which PRISMA and DESIS imagery have been acquired.

Finally, for mapping the above water biomass of floating-leaved and emerging aquatic vegetation we used the algorithm proposed by Villa et al. [68] previously applied to airborne APEX hyperspectral data on Mantua lakes and valid for a biomass range of 0–0.5 kg m$^{-2}$. Briefly, the algorithm was built on a band ratio in the red-edge and near-infrared APEX wavelengths and, due to a similar hyperspectral band setting, it can be easily adapted to both PRISMA and DESIS data.

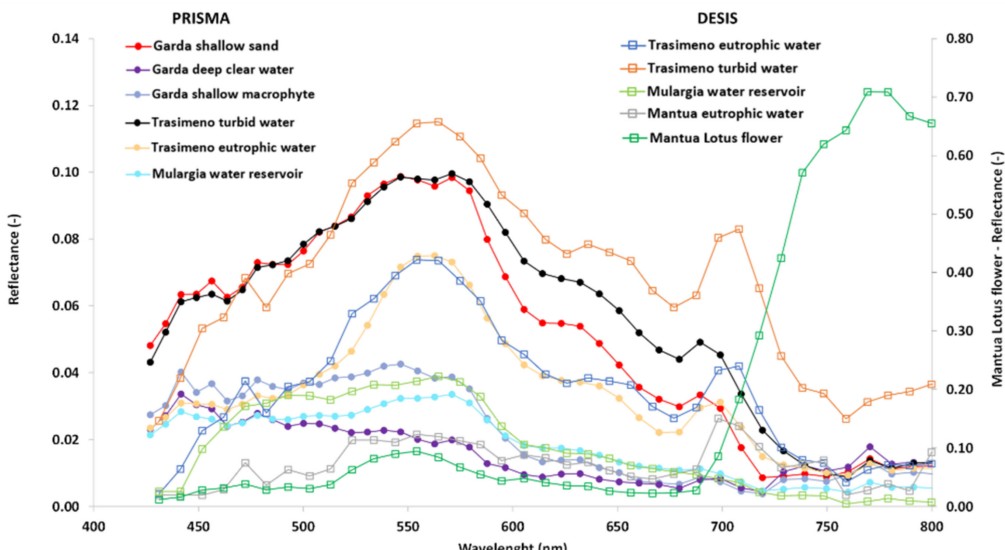

**Figure 2.** PRISMA and DESIS water reflectance. The spectra have been extracted to demonstrate the variability of hyperspectral reflectance across the water systems, from shallow to deep, and from clear to eutrophic or turbid and of aquatic vegetation (i.e., lotus flower) collected in the case study lakes investigated (Mantua, Mulargia, Trasimeno, and Garda lakes).

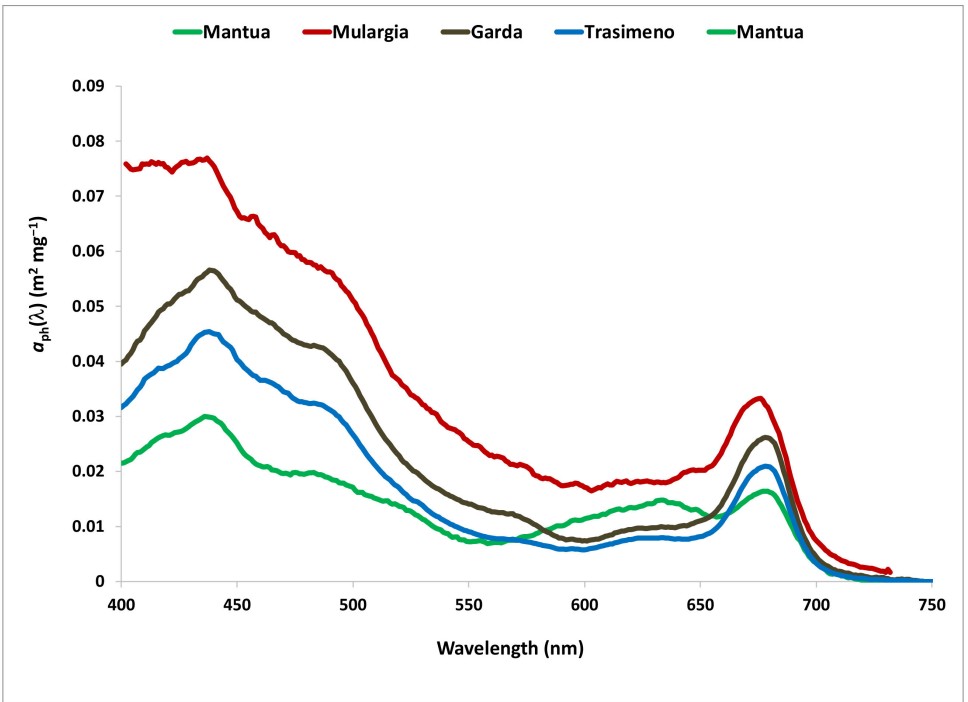

**Figure 3.** The specific absorption spectra of phytoplankton for the study areas used to calibrate the bio-optical model implemented in BOMBER.

**Table 2.** Calibration parameters of the bio-optical model used in BOMBER for estimating water quality parameters from PRISMA and DESIS imagery.

| Parameter | Garda | Mantua | Mulargia | Trasimeno |
|---|---|---|---|---|
| Spectral slope coefficient of CDOM absorption | 0.021 | 0.014 | 0.018 | 0.016 |
| Specific absorption of NAP at 440 nm | 0.05 | 0.30 | 0.11 | 0.20 |
| Spectral slope coefficient of NAP absorption | 0.012 | 0.008 | 0.012 | 0.013 |
| Backscattering exponent of TSM | 0.76 | 0.80 | 0.71 | 0.65 |
| Specific back-scattering coefficient of TSM at 555 nm | 0.0071 | 0.0111 | 0.0105 | 0.0119 |

## 4. Exploitation of PRISMA and DESIS Products for Water Quality Mapping

In this section we show how the PRISMA and DESIS hyperspectral images can generate aquatic ecosystems products for different types of water bodies. In particular, among the topics of greatest interest for lakes is the need to have spatial information on the distribution of primary producers (both phytoplankton and emerged and submerged aquatic macrophytes) and their temporal variability. Another key requirement for monitoring water quality is to have information relating to the transparency of the water, which in lakes with low humic substances, is directly related to the presence of total suspended solids in the water column. For the purposes of this review we limit the discussion of the products to the use of hyperspectral images for ecological studies, monitoring and management of aquatic environments.

### 4.1. Mantua Lakes

The fluvial-lake system of Mantua is located in Northern Italy (45°10′ N, 10°47′ E; surface area 6.2 km$^2$, mean depth 3.5 m; Figure 1) and was created in the 12th century by the damming of the Mincio River. The system is composed of three shallow lakes, Superior, Middle and Inferior lakes. In Superior Lake, the Vasarone sluice gate (dating back to 1190 A.C.) and the recently-opened Vasarina gate (2015) are regulated to produce electricity and maintain a constant water level in the three basins (17.5 m a.s.l. on the Superior Lake, and 14.5 m a.s.l. on the Middle and Inferior lakes) [69]. Elevated nutrient loads enter the fluvial-lake system from its main tributary, the Mincio River, that drains a watershed intensively exploited by agricultural and animal farming activities [70,71]. Therefore, this fluvial-lake system is highly productive with the coexistence of dense phytoplankton and macrophyte (emergent, submersed, floating-leaved and free-floating) communities [68,72]. Water is characterized by low transparency (Secchi disk depth < 1 m in summer) and high chlorophyll-a (Chl-a) concentration (up to 200 mg m$^{-3}$) [73,74]. The Mantua Lakes are part of the Mincio Regional Park, and since 2008 they have been part of the UNESCO World Cultural Heritage. Active management of the water and macrophytes is needed to ensure the conservation of the area, and enable the recreational and touristic activities developed on the lakes. Floating-leaved macrophytes (such as Nelumbo nucifera, Trapa natans and Ludwigia hexapetala) are kept under control by cutting and occasionally harvesting to avoid an excessive accumulation of organic matter on the bottom of the lakes that can lead to rapid infilling and the risk of hypoxia/anoxia events [68,75]. For a decade this study area has been investigated with different techniques which includes proximal and remote sensing techniques. For example, for the first application, Bresciani et al. [73] assessed short (intra-day) and medium (from weekly to monthly) term Chl-a dynamics by means of continuous spectroradiometric measurements, and Pinardi et al. [76] integrated the use of high frequency in situ measurements and multispectral satellite data to explore temporal (diurnal and seasonal) dynamics and spatial distribution of Chl-a concentration, together with physico-chemical water parameters. For the second application, various multispectral satellite (e.g., Landsat constellation, Sentinel-2, SPOT5) or hyperspectral sensors (e.g., APEX, MIVIS) were used to retrieve water quality [77], macrophyte products [68,78,79] or both [72,74,80] to assess the Mantua lakes system ecology at different spatial–temporal

scales and to support management plans and actions. In this small fluvial-lake where there are multiple pressures and diverse uses of the water, hyperspectral data can guarantee spatial and temporal information suitable for the study of primary producers, including the identification of the different functional groups and their seasonal evolution.

In order to provide relative information of the different primary producers present in the Mantua Lakes we used two DESIS images (2019/06/21 and 2020/06/22), the products of which are shown in Figure 4. Chl-a and Phycocyanin (PC) concentrations were estimated by means of BOMBER as further described also in Giardino et al. [27], with specific optical properties of the waters of the Mantua Lakes (see Table 2 and Figure 3), and previously applied to airborne hyperspectral APEX images. The comparison of the products with field data indicated good agreement: for the 21 June 2019 the Chl-a measured in situ via fluorimetric probe in the Middle and Inferior lakes [76] had a value of 20.0 and 15.4 mg m$^{-3}$ respectively versus satellite products of 21.9 and 16.0 mg m$^{-3}$; for the 22 June 2020 the in situ Chl-a was 23.5 and 17.7 mg m$^{-3}$ in the Middle and Inferior lakes respectively, versus satellite products of 25.0 and 16.8 mg m$^{-3}$ (Figure 4).

The possibility of having simultaneously spatial information related to the concentrations of the different phytoplankton photosynthetic pigments provides a high added value in the understanding of the quality state of aquatic environments. In the Chl-a maps a clear increasing gradient down the system from the Superior (median value of 13.65 and 16.08 mg m$^{-3}$ for 2019 and 2020 images, respectively) to the Middle Lake (median value of 16.80 and 26.35 mg m$^{-3}$ for 2019 and 2020 images, respectively) is shown. Notwithstanding the fact that portions of the Superior Lake had higher concentration due to water circulation driven by water discharge and by the presence of macrophyte meadows. Similarly, the lower average concentrations of PC in the Superior Lake on 22 June 2020 (median value of 10.98 mg m$^{-3}$) were due to its fluvial-lacustrine morphology which favors a shorter residence time compared to the Middle and Inferior lakes, in which the reduced current velocity of the waters facilitates the high growth of cyanobacteria that become dominant (for both lakes a median value ~26 mg m$^{-3}$).

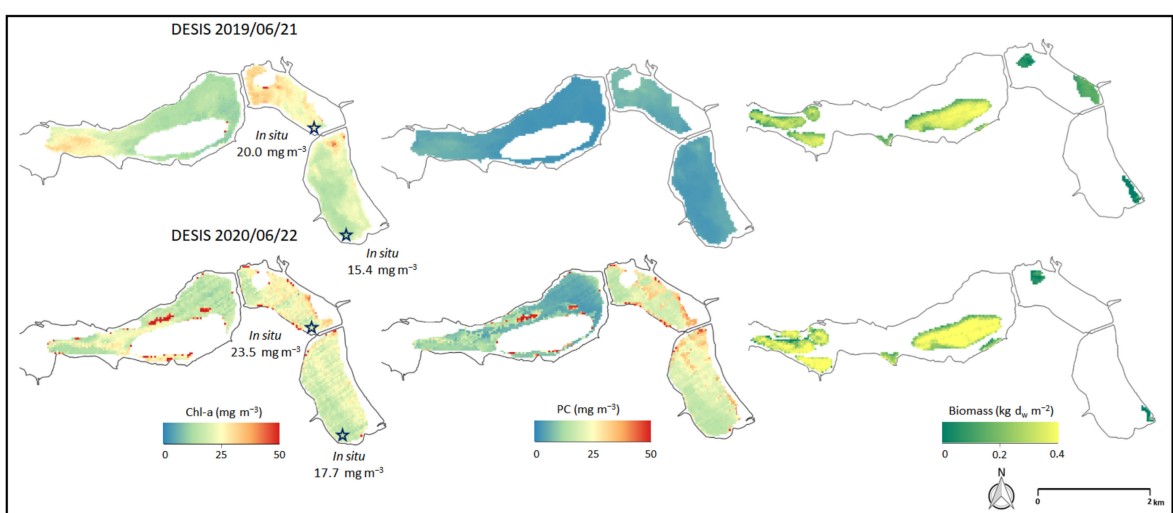

**Figure 4.** Maps of concentration of phytoplankton pigments (Chl-a and PC) and of floating-leaved and emergent macrophyte biomass obtained from DESIS images acquired on 21 June 2019 and on 22 June 2020. Blue stars (for Chl-a) are the locations of the in situ measurements with fluorimetric probes.

### 4.2. Lake Mulargia

On the Flumendosa river, located in Sardinia (Italy; 39°38′ N, 9°14′ E; Figure 1) a dam was built creating the Mulargia reservoir. Lake Mulargia has a surface area of 12 km$^2$ and a volume capacity of 347 hm$^3$, and it serves as a drinking water source (annually ~100 hm$^3$) for 700.000 inhabitants including the town of Cagliari and its hinterland as well as another

20 villages nearby. Soil erosion and nutrient loads (mainly phosphorous) from agricultural land can affect lake water quality leading to low transparency (elevated suspended solids) and high conductivity, with episodic algal bloom events resulting from the higher phosphorus loads. For this reservoir water temperature and water quality products, such as Chl-a, turbidity, transparency and a trophic status index, were mapped by means of multispectral satellite data (i.e., Landsat-7 and 8, Sentinel-2) for the period from 2013 to 2018 [17]. As for the previous case study, for these small waterbodies the use of hyperspectral imagery can provide added value as it allows the detection and quantification of algal blooms (such as cyanobacteria bloom) and the origin of suspended material (separation between the organic and inorganic) in the water column, supporting a warning system that is of crucial importance for a reservoir dedicated to drinking water supply.

The water quality maps retrieved from PRISMA (Figure 5, top) and DESIS (Figure 5, bottom), by means of the BOMBER code and related calibration (see Table 2 and Figure 3) describe the Chl-a, TSM and Inorganic Suspended Materials (Inorganic SM) concentrations (only produced with DESIS) with a range between 0 and 10 mg Chl-a $m^{-3}$ or g TSM (Inorganic SM) $m^{-3}$, and CDOM concentrations within a range from 0 to 0.7 $m^{-1}$. All the parameters were higher in August compared to the July situation, and looking at the spatial distribution of water quality parameters, the highest concentrations (median value: Chl-a 4.51 and 6.29 mg $m^{-3}$, CDOM 0.16 and 0.47 $m^{-1}$ and TSM 2.35 and 4.32 g $m^{-3}$, in July and August, respectively) were found in the eastern portion of the lake probably due to lower water circulation. This may result from the fact that a tributary is located in the western portion of the lake and the dam is in the south. Another area was the Chl-a concentration was high was the western portion of the reservoir close to the tributary inflow, probably due to nutrient enriched waters that favor phytoplankton blooms. The amount of yellow substances, indicated by CDOM, were significantly higher in August compared to the previous image in July (+70%). The inorganic component of the suspended material was about 66% of the total suspend matter in the map produced from the mid-August DESIS image. These findings are comparable to the results reported in Bresciani et al. [17].

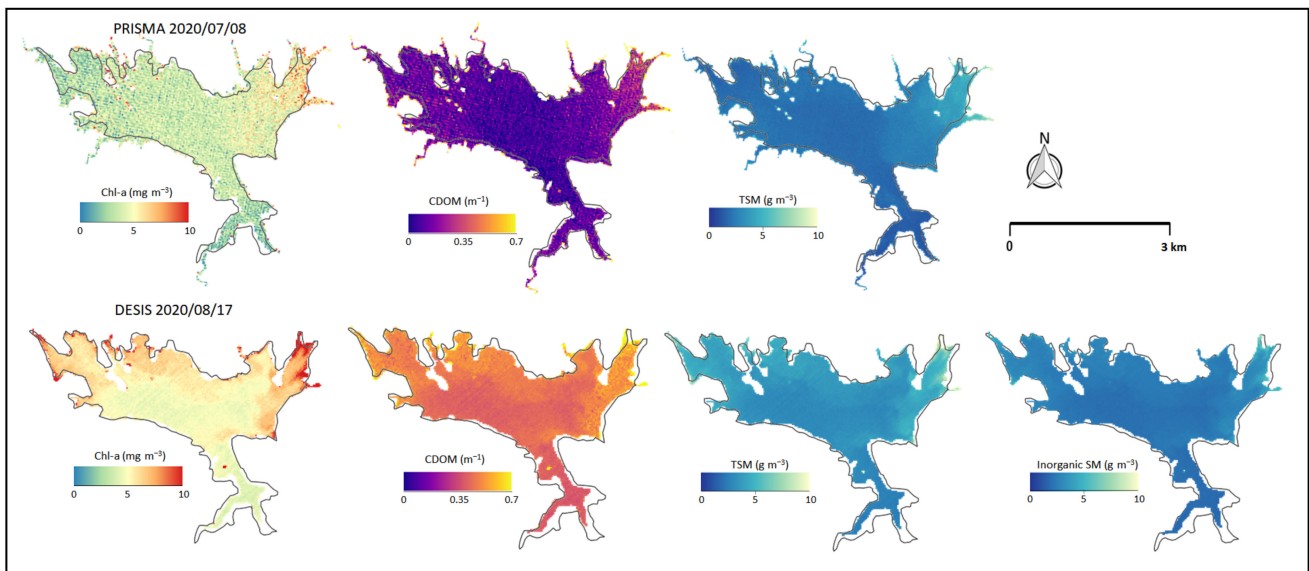

**Figure 5.** (**Top**) Maps of concentration of Chl-a, CDOM and TSM obtained from PRISMA image acquired on 8 July 2020. (**Bottom**) Maps of concentration of Chl-a, CDOM, TSM and Inorganic SM obtained from DESIS image acquired on 17 August 2020.

### 4.3. Lake Trasimeno

Lake Trasimeno is located in Central Italy (43°08′ N; 12°06′ E; surface area 120.5 km²; Figure 1) and is characterized by shallow (maximum depth 6.3 m) and turbid (Secchi disk 1.1 m, and SPM 10 g $m^{-3}$ for the period 2002–2008) waters. Precipitation is the main

driver of lake water level as the lake is only fed by small watercourses and is essentially endorheic, having no natural outlet [81]. The water column is unstratified due to wind action which also favors sediment resuspension [81]. The lake's trophic status is eutrophic with Chl-a concentration up to 90 mg m$^{-3}$ [82]. The primary producers are phytoplankton (e.g., chlorophytes and dinoflagellates) including cyanobacteria (e.g., *Cylindrospermopsis raciborskii*, *Planktothrix agardhii*), and emergent and submerged macrophytes located in littoral zones and in an open bay in the south-east portion of the lake (Oasi La Valle), respectively [83,84]. Agriculture and animal farming are the main pressures in the watershed, while fishing, tourism and recreational activities exert pressure directly on the lake. Lake Trasimeno is in fact an area of exceptional value for flora and fauna richness and for species biodiversity and it is part of a Natural Regional Park and of two Natura 2000 sites (IT5210018 and IT5210070). It is important in such kind of waterbodies to characterize the diurnal variation of phytoplankton abundance, by means of a proxy such as Chl-a concentration, because algal blooms can grow and dissipate within a few days [85]. In addition, the seasonal phytoplankton phenology and distribution are influenced by this short-term variability [86]. Hyperspectral sensors allow the retrieval of Chl-a and secondary pigments concentration together with other water quality parameters, and the use of continuous in situ spectroradiometers allow the capture of hourly variation of phytoplankton abundance (e.g., [87]). Merging high-frequency data with satellite observations allows one to go beyond the limited spatial coverage of in situ sampling [88], and provide spatial and temporal information for water monitoring and management [89]. Several studies using remote sensing techniques have been performed in this case study area. For example, the physical parameter surface water temperature and Chl-a concentration were both retrieved by satellite data (MODIS and MERIS) and they were positively correlated in the period from the spring to the end of summer (2005–2008) when biological activity was at its highest [90]. Recently, a study on Lake Trasimeno analyzed the integration of detailed spatial information from new generation remote sensing satellites (Sentinel 3-OLCI and Sentinel 2-MSI) with high frequency observations from in situ optical sensors (WISPstation) to improve water quality monitoring and identify drivers of Chl-a dynamics [84]. Giardino et al. [63] used a hyperspectral MIVIS image acquired from an airborne sensor to provide simultaneous maps of the concentration of the optically active parameters, water column heights and benthic substrate types. Recently Niroumand-Jadidi et al. [91] retrieved water quality maps from PRISMA images and the results suggested the high potential of PRISMA imagery in mapping water quality parameters in Lake Trasimeno.

For the estimation of the concentrations of Chl-a and TSM, the BOMBER code was still adopted with related site-specific parameters (see Table 2 and Figure 3), and as also previously applied to airborne hyperspectral MIVIS imagery. For this specific example the products were evaluated based on five match-ups of hyperspectral products (3 × 3 pixel boxes) and in situ data: for Chl-a and TSM with values of root mean square error of 11.1% and 15.8%, respectively, and mean absolute error of 0.94 mg m$^{-3}$ and 1.58 g m$^{-3}$, respectively.

In Figure 6 is shown a series of maps retrieved in different seasons, which highlights the potential of using hyperspectral images for temporal analysis. In fact, the combination of different sensors can improve the temporal analysis useful to understand, for example, phytoplankton dynamics. Looking at the evolution of Chl-a concentration, there is a clear seasonal pattern with lower values in late spring (median Chl-a of 1.63 and 2.65 mg m$^{-3}$ on 18 June 2019 and 26 May 2020, respectively) and progressively higher values from summer to late summer (median Chl-a value of 5.45, 8.77 and 31.54 mg m$^{-3}$ in July, August and September, respectively). This result is in accordance with late summer phytoplankton blooms reported in previous work [84]. The same temporal pattern was found for the total suspended matter, which can reduce the penetration of light in the water column in some portions of the lake.

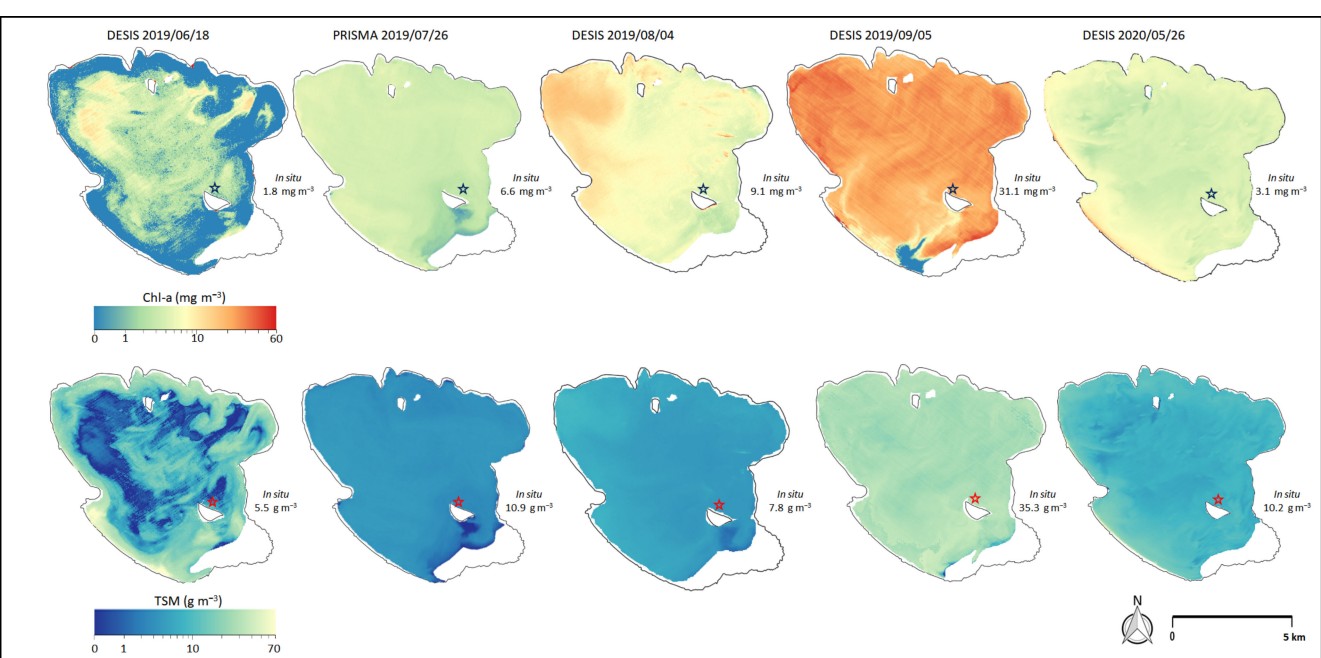

**Figure 6.** Maps of Chl-a (**top**) and TSM (**bottom**) concentration from DESIS and PRISMA images. Blue (for Chl-a) and red (for TSM) stars are the location of the in situ measurements.

## 5. Lake Garda

Lake Garda is the largest Italian lake and is located in the subalpine region (45°40′ N, 10°41′ E; surface area 368 km², water volume 49 km³; Figure 1). In 1951 the Salionze dam was completed near the headwaters of the lake emissary (Mincio River), allowing the management of downstream water needs (e.g., crop irrigation) and prevent flooding through water level regulation [92]. As with the other deep subalpine lakes of the same region, the Lake Garda represents a strategic water resources for drinking, agriculture, industry, energy, fishing and tourism [62,93]. Lake Garda is classified as oligo-mesotrophic (mean Chl-a 3 mg m$^{-3}$; Secchi disk depth 5 m in summer and 15 m in late winter [94]) with two different morphological areas: the largest sub-basin extending from north to southwest area, characterized by the deepest depths, and the south-eastern shallower sub-basin [62]. In the south portion of the lake the littoral zone and the shallower water are characterized by the presence of emergent (e.g., *Phragmites australis*) and submerged macrophytes (e.g., *Chara globularis*, *Vallisneria spiralis*, *Lagarosiphon major*) [95]. Retrospective analysis of lake bottom and macrophyte meadows in the littoral zone around the Sirmione Peninsula (Figure 1) was performed for the period from 1997 to 2005 [96], and until 2010 [95] by means of MIVIS hyperspectral airborne images. Such studies allow the identification of the loss of well-structured submerged vegetation and its replacement by sparsely vegetated or unvegetated substrates, with repercussions on the sustainable use and conservation of natural lacustrine resources [95]. Remote sensing techniques have been used for a long time in Lake Garda to examine water parameters (from surface temperature to Chl-a and turbidity). To our knowledge the first study published on the retrieval of Chl-a concentration from multispectral data (Landsat TM images) was published by Brivio et al. [97]. Moreover, several methodological papers followed, with the use for example of the Hyperion image to retrieve Chl-a and other water quality parameters [98], or the use of MODIS and MERIS multispectral sensors to investigate relationships between lake surface temperature and Chl-a concentration in the shallower zone of the south portion of Lake Garda [90]. Applied studies were also performed to support water monitoring and management in accordance with the Water Framework Directive (e.g., [99]), to evaluate the influence of dust transport in phytoplankton growth [100], and to quantify the turbidity due to the discharge of the Adige-Garda spillway on Lake Garda [101].

As an example, from the PRISMA image of the 2020/09/10 different products of water quality and of lake bottom retrieved with BOMBER and related calibration (see Table 2 and Figure 3) are reported in Figure 7. The maps show the Chl-a and TSM concentrations of the south-west portion of the basin, which show also a good agreement with the in-situ data. The Chl-a and TSM concentrations were low on this date confirming the oligo-mesotrophic status of the waters, with higher values up to 3.05 mg Chl-a m$^{-3}$ and 3.33 g TSM m$^{-3}$ in the southern littoral portion of the study area. The map of the Sirmione Peninsula colonized by sparse and dense vegetation is also shown in Figure 7. This estimate of the macrophyte cover of the shallower portion of the Peninsula is in accordance with the results for September 2010 published by Bresciani et al. [95] that reported a reduction of the density of the macrophyte meadows compared to the previous decade. For the water quality products, the Chl-a concentration estimated from the PRISMA image was 1.30 mg m$^{-3}$ with respect to 1.40 mg m$^{-3}$ measured in situ, and the TSM concentration was 0.43 g m$^{-3}$ compared to 0.50 g m$^{-3}$ measured in situ (Figure 7).

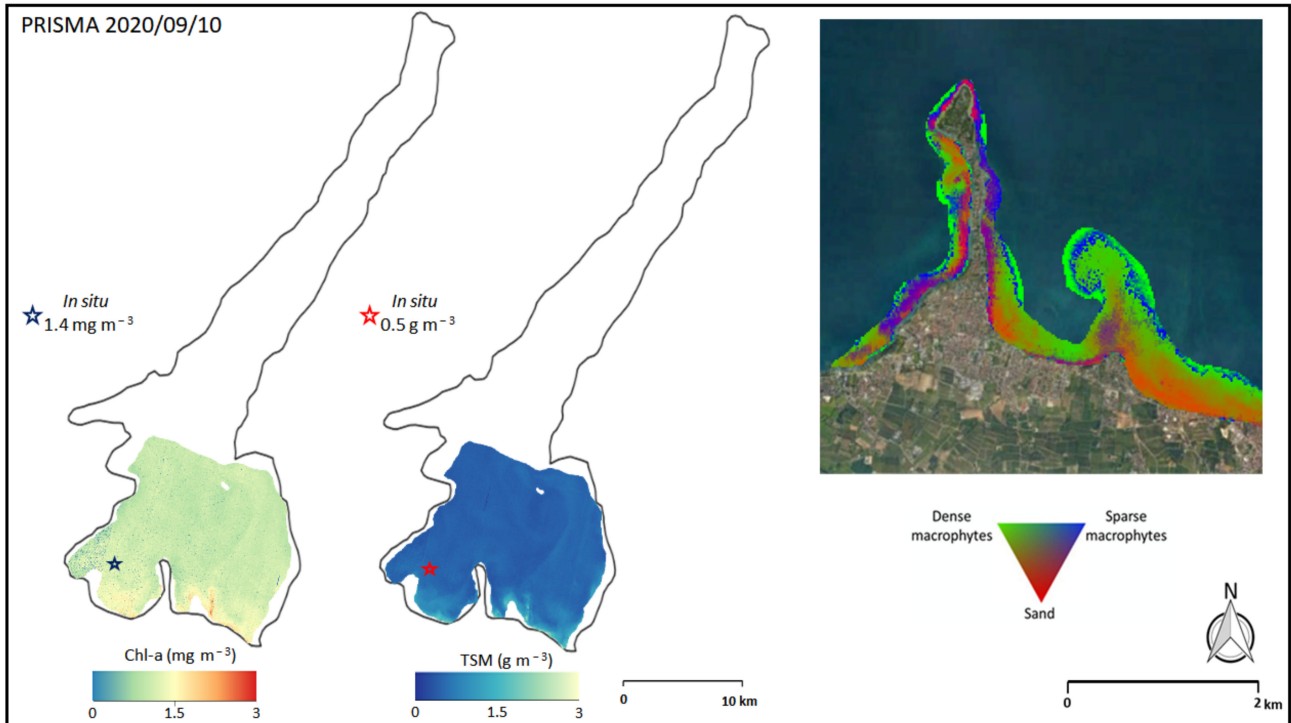

**Figure 7.** Maps of Chl-a (**left**) and TSM (**center**) concentration, and (**right**) of the bottom coverage of the Sirmione Peninsula in the south portion of Lake Garda (see also Figure 1 for the location of this area) retrieved by PRISMA image acquired on 10 September 2020.

## 6. Conclusions

Given the state-of-the-art algorithms and the improved EO technology, both multispectral sensors and ocean color radiometers are currently being used for a variety of purposes such as monitoring frameworks from local to regional scales (e.g., [10,39]), while the first set of satellite products for evaluating the effects of climate change on lakes has been also released [14]. Along with such developments, the field of aquatic hyperspectral remote sensing is also advancing rapidly and it is a fact that about 20 reviews have been published in the last four years on the subject [31] and references herein. Several spaceborne imaging sensors have been launched in recent years while in the near future new missions are planned. To focus on satellite sensors which provide a spatial resolution between ~17 and ~33 m, as recommended by Dekker [102], missions such as PRISMA [50], DESIS [103] or GaoFeng-5 [104,105] have to be mentioned as currently orbiting, while EnMAP [106], PRISMA-SG, CHIME and SBG [107] will follow in the next years.

The study presented in this review aimed to demonstrate some examples of water quality mapping based on PRISMA and DESIS imagery of four diverse Italian water systems supporting a variety of ecosystems services, from recreation to drinking water supply. The examples provided include the simultaneous retrieval of Chl-a, TSM, CDOM, the mapping of fractional cover of bottom types, the assessment of phytoplankton pigments as well as of emergent macrophytes biomass. For some of these parameters (e.g., TSM) multispectral sensors are already providing valuable products, while further studies [57] are also using data from the same missions for advancing the estimation of water quality parameters.

While a diversity of case studies was presented on the use of hyperspectral imagery indicating the enormous scientific potential of this approach, the benefits of its use are likely to be case-specific. A favorable cost–benefit analysis often results in situations where there are high numbers of recreational users, health implications or where it is important to manage reputation, for example to maintain confidence in a water supply [108]. Many of the examples presented here certainly represent such high value situations for tourism and recreation (Garda, Trasimeno, Mantua) or water supply (Mulargia). In particular Lake Garda was the second most visited tourist destination in Italy, with 24.6 million overnight stays in 2017 with environmental preservation and natural heritage ranking among six attributes in determining visitor satisfaction [109].

To conclude, the unique value of spaceborne imaging spectroscopy has to be considered in synergy with all classes of sensors as cooperation among missions remains the key approach to study aquatic ecosystems [110], and to monitor water quality by also addressing operational needs associated with policy obligations [111]. To this aim, some challenges in exploiting imaging spectroscopy are open and would require further research efforts. Among the others we would like to mention is the need of expanding hyperspectral validation sites and training the user community. This would enable the integration of datasets and amplify the value of information strands too often left separate.

**Author Contributions:** Conceptualization, M.B., C.G. and M.P.; data curation, M.B., M.P., A.F., A.P. and S.M.; methodology, M.B., C.G. and M.P.; validation, M.B., A.F., A.P., S.M. and C.G.; supervision, M.B. and C.G.; writing—original draft, M.B., C.G., G.F. and M.P.; writing—review and editing, M.B., C.G., G.F. and M.P. All authors have read and agreed to the published version of the manuscript.

**Funding:** Funding for satellite processing activities and data analysis was supported by the EU Horizon 2020 programme with the projects PrimeWater (GA No. 870497) and Water-ForCE (GA No. 101004186) and by the ASI-PRISCAV project (GA No. 2019-5-HH.0).

**Institutional Review Board Statement:** Not applicable.

**Informed Consent Statement:** Not applicable.

**Data Availability Statement:** The datasets generated during and/or analyzed during the current study are available from the corresponding author on reasonable request.

**Acknowledgments:** Part of this work was supported by EU Horizon 2020 programme with the project EOMORES (GA No. 730066) and of ESA-CHIME (4000125506/18/NL/IA). PRISMA and DESIS date were used upon agreement of respective licenses to use. We are very grateful to E. Lopinto from ASI and to N. Pinnel and U. Heiden from DLR for relevant discussion on PRISMA and DESIS missions. We would like to thank the three anonymous reviewers for comments that improved this manuscript.

**Conflicts of Interest:** The authors declare no conflict of interest.

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
