# Peer review of "Application of New Hyperspectral Sensors in the Remote Sensing of Aquatic Ecosystem Health: Exploiting PRISMA and DESIS for Four Italian Lakes"

_resources, doi:10.3390/resources11020008_

Round 1
Reviewer 1 Report
The article presents various implementations of hyperspectral data recorded by two state-of-the-art PRISMA and DESIS sensors. It is a valuable and compact review of the knowledge about remote sensing of the Italian Lakes. Results are presented for four locations differing by optical water properties. Each location is described by varying parameter set. The story goes smoothly from one lake to another. Language is reader-friendly. However, the variety of measures used, combined with jumping between the lakes, makes it hard to follow parameters and introduces some doubts about the algorithms used, e.g. are the Chl-a and TSM algorithms the same for Lake Garda as the Lake Trasimeno? I would expect a few more sentences describing each of the method/measure used, especially when examples were computed (like in Figure 6). I believe that would introduce some order and would prevent reader from getting lost.
I also miss in conclusions a summary of the added value of using hyperspectral sensors and their advantages in relation to multispectral sensors. It is partly mentioned in lines 104-106, 217-218, 316 - 325 and referenced (position 38), but I think it should be clearly stated in conclusions, as it is an important aspect of the whole work.
I have also some specific comments relating to the overall impression presented above:
- line 63 - I am surprised to see citation of initiatives in Latin America, but no reference to European Water Frame Directive and application of remote sensing in that context. It would be valuable to give some more information about remote sensing systems of the lakes in Europe. In particular, when there is a short glimpse of WFD (position 88) at line 386.
- 103 - 109 - It would be good to have here some references and at least few words about APEX and MIVIS sensors that you refer to further in the text, instead of GaoFen-5 or HISUI that do not appear later on. Are they so similar to PRISMA and DESIS that you may apply the same algorithms?
- line 159 - 160- After a sentence like this "while further research efforts to improve the retrieval of water reflectance are still under development (51)." I would expect Figure 2 to be a comparison of satellite data with e.g. the WISP spectroradiometer data mentioned in Figure 5. At least for some spectra.
- 230 - 232 -I would expect more information about methods / parameters/ measures and their limitations used to map macrophyte biomass. It is a parameter differing from the other analysed in this work. Based on the Figure 2 and short notes further in lines 247-249 it may be inferred that it is only biomass of floating on the surface Lotus leaves, but it should be more precisely written how to interpret the "macrophyte biomass". How was it derived (was it based on the dry biomass parameter)?
- Line 271 - 272: It seems that all results for Lake Mulargia are based on the algorithms implemented in BOMBER software. It would be good to have at least short description of the model and the assumptions for that Lake, as the results seem to depend to a great level on the assumptions adopted.
- 340 - 342 – I would appreciate it if all algorithms were cited with errors stated like here. I mean Chl-a and PC retrievals mentioned in lines 219 - 229 or dry biomass estimation in line 230-232.
- line 280 - 287 - How those results relate to OLCI and MSI images, and other studies? It was nicely done for the next lake in lines: 350 - 351.
- 394 - 395, 399 - How were those values derived? Were those the same algorithms as used before? Did they work also in oligothrophic waters of Lake Garda?
- 391 - There is only an extremely brief reference to the retrieval of bottom vegetation and it appears here out of the blue. Please extend that part and refer to it in the introduction, especially as it is maybe connected, but it is not a water quality parameter.
Nevertheless, those are only minor comments and the work may be published after subtle changes.
Author Response
Many thanks for your comments, we appreciate. We upolad a text file with the specific answers.

Reviewer 2 Report
This useful review article places the new hyperspectral sensors PRISMA and DESIS in the context of water quality management. The paper provides nice examples of PRISMA and DESIS water colour products of Italian lakes, as well as useful bibliography, which will be valuable for those interested in this topic.
Overall: I missed one paragraph on the challenges of working with hyperspectral satellite data which are several, from the very few hyperspectral validation sites available to a not trained user community.
A few small recommendations follow:
Line 39: keep the same style of writing “in situ”, in italic? Please, check other places in the text too as line 47.
Lines 69-70: instead of (…Sentinel-2) to coarse (e.g., ENVISAT-MERIS, Sentinel-3 OLCI) ïƒ (…Sentinel-2-MSI) to coarse (e.g., ENVISAT-MERIS, Sentinel-3-OLCI)
Line 74: remove „maps of„
Line 115: in Italy.
Line 123: what’s the spectral sampling of PRISMA?
Line 156: did you use DESIS binned data? Any flag applied to the atmospheric corrected PRISMA and DESIS products?
Figure 2: The figure is nice, but it is very difficult to differentiate between DESIS and PRISMA. I would suggest to show 3 plots, one only with spectrum from PRISMA (Garda), another one DESIS (Mantua) and the third one with coincident DESIS and PRISMA spectrum (e.g. Transimeno and Mulargia). In the last one, you could apply the same color for both sensors, but using different symbols.
Lines 266-269: multispectral sensors also allow the detection of algal blooms. And how can hyperspectral and multispectral data be used to investigate the origin of suspended material? You should be more specific here to show the added value of hyperspectral sensors.
Lines 339-341: it would be nice to have one sentence on how the match-ups were extracted: how many points, temporal window (time of the overflight?) and spatial average (3 by 3 pixel box?)?
Figure 5: first DESIS map: the spatial distribution looks strange compared to the other plots. Any idea why?
Line 379: remove the comma.
Lines 417-420: the sentence is a bit hard to understand and needs reformulation.
Author Response

(The authors gave the same response as above.)

Reviewer 3 Report
The manuscript of Bresciani et al. compares the use of two orbital hyperspectral radiometric sensors to obtain water quality variables (chlorophyll-a, phycocyanin, turbidity, etc.) in four Italian lakes with distinct characteristics. The authors did a good job compiling images and in-situ results, and presenting maps showing the concentrations of these variables.
The manuscript is presented with methods, results and discussion merged in sections which are specific to each lake, which made it interesting to read but lacks essential details on the methods implemented to generate these figures. Details missing in the methods include the description of the models used to calculate all variables, with their specific equations and citations. This must be included in the manuscript before publication.
Regarding the title, the manuscript does not present an overview of water quality remote sensing methods applied in Italian Lakes, as informed in the title. It actually uses data from two orbital sensors to determine water quality indexes in four lakes in Italy. Therefore, I suggest the title is changed to better describe the content of this manuscript.
In some instances, uncertainty metrics were not used to compare radiometry-derived products with in-situ data, and the authors exposed both results in the text allowing for comparison. In other instances, uncertainty indexes were used. If used, these metrics must be described in the methods. In addition, it would be informative to see plots comparing observed and measured chlorophyll-a and the other products, for each of the two sensors, and each of the four lakes. Most importantly, the authors should standardize data description and comparison in the text (i.e. use of uncertainty metrics or direct comparison in the text). I strongly recommend some uncertainty metrics are used and this comparison between observed (in-situ) and predicted (radiometry-derived) data is shown in plots.
Finally, the manuscript’s conclusion is that hyperspectral data can be used to estimate water quality variables in lakes with different tropic states and geomorphologic characteristics. This has been shown in a variety of studies, cited in this manuscript. Considering that, the manuscript brings no new knowledge about this topic. In addition, I expected to see more discussion about the advantages of using hyperspectral images to obtain these products, rather than the current conventional multispectral satellites. Therefore, I suggest some applied goal is explored in the manuscript, so the potential of the generated images to describe the lakes’ trophic levels is highlighted.
Please find minor suggestions listed below:
Line 38: Replace “while it provide” with “while it provides”.
Line 40: Replace “In situ measurements” with “In-situ measurements”.
Line 41: Replace “phytoplankton counts” with “phytoplankton abundance”.
Line 47: Replace “In situ observations” with “In-situ observations”. Please do this replacement every time the expression “in-situ” is used as an adjective before a noun.
Line 48: In “to the to the possibility”, remove “to the”.
Line 68: Include a coma after “To date”.
Line 74: Remove “produced”.
Line 107: “A few years”, please include the s in ‘year’.
Line 131: Include a coma after “in this study”.
Figure 1. Please include a map of Italy to facilitate the visualization of the lakes’ locations.
Line 153: “the dates of” please include the “s” in “date”.
Lines 340-342: Uncertainty metrics are presented. These metrics must be described in the Methods section, with their specific equations and citations.
Author Response

(The authors gave the same response as above.)

Round 2
Reviewer 3 Report
The modifications included by the authors significantly improved the manuscript.